# Design of Novel Membranes for the Efficient Separation of Bee Alarm Pheromones in Portable Membrane Inlet Mass Spectrometric Systems

**DOI:** 10.3390/ijms25168599

**Published:** 2024-08-07

**Authors:** Stevan Armaković, Daria Ilić, Boris Brkić

**Affiliations:** 1University of Novi Sad, Faculty of Sciences, Department of Physics, Trg Dositeja Obradovića 4, 21000 Novi Sad, Serbia; 2BioSense Institute, University of Novi Sad, Dr Zorana Đinđića 1, 21000 Novi Sad, Serbia; daria.ilic@biosense.rs

**Keywords:** MIMS, membranes, bee alarm, pheromones, PDMS, PEG, GFN2-xTB, MD, DFT

## Abstract

Bee alarm pheromones are essential molecules that are present in beehives when some threats occur in the bee population. In this work, we have applied multilevel modeling techniques to understand molecular interactions between representative bee alarm pheromones and polymers such as polymethyl siloxane (PDMS), polyethylene glycol (PEG), and their blend. This study aimed to check how these interactions can be manipulated to enable efficient separation of bee alarm pheromones in portable membrane inlet mass spectrometric (MIMS) systems using new membranes. The study involved the application of powerful computational atomistic methods based on a combination of modern semiempirical (GFN2-xTB), first principles (DFT), and force-field calculations. As a fundamental work material for the separation of molecules, we considered the PDMS polymer, a well-known sorbent material known to be applicable for light polar molecules. To improve its applicability as a sorbent material for heavier polar molecules, we considered two main factors—temperature and the addition of PEG polymer. Additional insights into molecular interactions were obtained by studying intrinsic reactive properties and noncovalent interactions between bee alarm pheromones and PDMS and PEG polymer chains.

## 1. Introduction

Honeybees (*Apis mellifera*) play a crucial role in maintaining ecological balance and supporting agricultural productivity through their pollination activities. As primary pollinators, bees are integral to the reproduction of many flowering plants, contributing to biodiversity and the stability of ecosystems. Additionally, their pollination services are vital for the production of a significant portion of the world’s food crops [1]. Despite their importance, bees are increasingly threatened by a variety of factors, including diseases, habitat degradation, and rising environmental pollution. These challenges necessitate the continuous evolution and enhancement of monitoring techniques in beekeeping. To effectively manage and protect bee populations, it is imperative that monitoring systems are updated to incorporate advancements in technology, enabling accurate and real-time detection of changes within the hive environment [2,3]. Such improvements are essential to address natural fluctuations and mitigate the impacts of widespread environmental contaminants, ensuring the sustainability of bee populations and the ecosystems they support.

The profile of volatile organic compounds (VOCs) inside a beehive is a critical factor in modern beekeeping, impacting various aspects from disease management to apitherapy. The presence of specific VOCs in a beehive can act as biomarkers for various diseases. For example, certain compounds are released when bees are infected with pathogens like Nosema or Varroa mites (propanoic acid, acetamide, etc.). By regularly monitoring the VOC profile, beekeepers can detect these biomarkers early, enabling them to take timely actions to manage and treat diseases before they cause significant harm to the colony [4]. Furthermore, apitherapy, which involves the use of bee products and even the air from inside the hive for therapeutic purposes, relies on the purity of the hive environment. The air inside the hive should be free from pollutants and contaminants to ensure its efficacy and safety in apitherapy. By keeping track of the VOC profile, beekeepers can ensure that the hive air remains clean, thus maintaining the therapeutic quality of the air used in treatments.

The primary indicators of any alteration within the bee colony dynamics lie in the realm of bee semiochemicals, notably bee pheromones [5]. These compounds, constituting a fascinating subset of VOCs, play pivotal roles in mediating communication and regulating behavior within bee colonies. Bee pheromones, characterized by their chemical nature, are typically polar molecules with relatively high molecular weights. For instance, one notable example is the queen mandibular pheromone (QMP), a complex amalgamation of multiple compounds. QMP serves as a linchpin in upholding the social structure of the hive. Additionally, alarm pheromones such as 2-heptanone, Isopentyl acetate, (Z)-11-Eicosane-1-ol, and 1-Octanol, whose structures are presented in Figure 1, represent another critical category, swiftly alerting bees to potential threats, particularly during disease outbreaks.

The chemical properties inherent to these pheromones facilitate their detection and recognition by worker bees, thereby exerting profound influences on various aspects of colony behavior. These influences span from regulating foraging activities and swarming behaviors to ensuring meticulous care of the brood [6].

The characterization of VOC profiles in bee air is typically performed using sophisticated analytical techniques, including high-performance liquid chromatography (HPLC), mass spectrometry (MS), liquid chromatography-tandem mass spectrometry (LC-MS/MS), and gas chromatography-mass spectrometry (GC-MS) [6,7,8]. While these methods are highly accurate and capable of identifying a wide range of compounds, they have several drawbacks, such as being time-consuming, costly, requiring laboratory analysis, and lacking real-time tracking capabilities.

To overcome the limitations of traditional differential techniques and ensure precise analysis of VOCs within beehives, researchers are turning to membrane inlet mass spectrometry (MIMS). This technique is celebrated for its simplicity, accuracy, rapid detection capabilities, and high sensitivity, which allow for the detection of VOCs at low parts per trillion (ppt) levels without the need for sample preparation [9].

The operation of MIMS is straightforward and relies on pervaporation separation [10]. A thin polymer-based membrane serves as the interface between the air or liquid sample and the vacuum system of the mass spectrometer. This membrane selectively permits the passage of organic compounds while blocking water molecules. The analytes of interest dissolve into the membrane, diffuse through it, and then evaporate into the vacuum system. Once in the vacuum, the analytes enter the ion source for ionization and subsequently the mass analyzer for spectral analysis. The resulting mass spectrum is used to determine the concentration levels of the substances by analyzing the intensity of the spectral peaks [10,11].

Recently, a portable MIMS system specifically designed for analyzing beehive air VOCs has been validated. One of such systems realized and employed by our research group is presented in Figure 2. This system employs a polydimethylsiloxane (PDMS)-based membrane, which is selective for VOCs, along with electron impact (EI) ionization and a single quadrupole (Q) mass analyzer. This configuration supports both online and offline analysis, offering comprehensive scanning across a mass range of 0–300 m/z in a short analysis time of approximately 2–3 min. Additionally, it provides selected ion monitoring (SIM) with a very fast response time [12]. In the area of volatile organic compound (VOC) analysis, the application of spectrometry MIMS has drastically improved detection capabilities. However, while MIMS offers an abundance of advantages, including rapid and sensitive detection without sample preparation, its reliance on a PDMS membrane presents a unique challenge in the analysis of polar bee pheromones.

Polar VOCs, like alcohols, ketones, and acids, have functional groups (e.g., hydroxyl, carbonyl, and carboxyl) that create uneven charge distributions, making them soluble in polar solvents like water and able to form hydrogen bonds. These properties result in higher boiling points and reactivity in various chemical processes. Non-polar VOCs, on the other hand, such as alkanes, alkenes, and aromatic hydrocarbons, have uniform charge distributions due to similar electronegativity among their atoms. They are insoluble in water but soluble in non-polar solvents like oils, with lower boiling points [13,14]. These compounds are commonly utilized in fuels, solvents, and various industrial applications owing to their volatility and energy content, and they can be detected within the hive as pollutants, including polycyclic aromatic hydrocarbons (PAHs); polychlorinated biphenyls (PCBs); and benzene, toluene, ethylbenzene, and xylene (BTEX) compounds [12]. Additionally, the MIMS detection technique using a membrane probe with a PDMS membrane was also utilized in the study [9]. The target breath VOCs were two polar and two non-polar compounds (acetone, ethanol, isoprene, and n-pentane) with molecular masses below 100 Da. Considering the diversity in the nature of the experiments and the detection of different VOCs with varying characteristics, one can observe distinct interaction dynamics between the PDMS membrane and specific compounds [15].

Furthermore, during in-field air sampling from beehives using MIMS [12], the PDMS membrane—a crucial component of the membrane probe—functions as a selective barrier. This membrane permits only non-polar or low-mass polar compounds to permeate into the vacuum system for analysis. This selective permeability is beneficial for many VOCs but poses a significant hurdle when attempting to detect polar bee pheromone molecules. These crucial chemical signals, often composed of polar functional groups, face difficulty in traversing the PDMS membrane due to their inherent affinity for water and their higher polarity. As a result, the exclusion of polar bee pheromones from the vacuum system diminishes the ability of MIMS to accurately capture and analyze these vital compounds. This limitation hampers our understanding of bee communication and behavior, as pheromones play a fundamental role in colony cohesion, reproduction, and defense. For future in situ experiments for air analysis from beehives, it is crucial to investigate the interactions between the target VOCs and the PDMS membrane to ensure accurate detection and measurement. Additionally, exploring potential enhancements and modifications of the membrane is essential for improving the efficiency of the membrane for identifying polar molecules with higher masses. This will greatly increase the number of target compounds that can be detected in beehive atmospheres using MIMS, particularly bee pheromones, which are mainly polar molecules.

Atomistic calculations are crucial in computational science, providing detailed insights into the atomic-level behavior of matter. The two most frequently applied atomistic calculations for predicting molecular properties are quantum mechanical calculations based on the density functional theory (DFT) approach [16,17,18,19] and molecular dynamics (MD) simulations based on force fields [20,21,22,23]. In efforts to achieve the accuracy of quantum mechanical calculations and the speed of force-field calculations, a set of outstanding semiempirical quantum mechanical methods have been developed [24]. All of the mentioned approaches have their own sets of pros and cons, but when combined together, they offer a powerful tool in predicting the various molecular properties and are essential for expanding our understanding and driving innovation across numerous scientific and technological fields. By predicting material properties and behaviors before synthesis or production, they save valuable time and resources [25,26,27,28]. When integrated with experimental data, atomistic calculations offer a complete understanding of various phenomena, making them a foundational tool for studying new molecules and materials [29,30,31,32,33,34]. Researchers have access to a wide range of theoretical methods, allowing them to explore atomic mechanisms with remarkable flexibility [35,36,37,38,39].

In this work, we applied a range of computational methods to investigate how interactions between heavier polar bee alarm pheromones and polymer materials can be manipulated at a molecular level. The main challenge in using the well-known PDMS polymer is that, while it is suitable for polar molecules, it is only effective for lighter molecules. Due to strong interactions with heavier polar molecules, PDMS traps these molecules, preventing their separation and thus making it unsuitable for use in mass spectrometric systems for these purposes.

Our essential task in this study was to reduce the interaction strength between heavier polar bee pheromone molecules and PDMS, which would enable the application of this widely available material as an inlet for aforementioned molecules in gas separation systems. To reduce the interaction and to make the material more permeable for heavier polar bee pheromone molecules, we considered two factors: temperature and blending with the PEG polymer.

Temperature is an easily controllable parameter, and in portable MIMS systems, it can be readily adjusted from room temperature up to 100 °C. Increasing the operating temperature makes the polymer chains more flexible, enhancing the permeability and separation of larger and polar molecules. The influence of this crucial parameter has been investigated through molecular dynamics (MD) simulations conducted in this study.

Due to the presence of methyl groups, PDMS is hydrophobic and, therefore, compatible with non-polar molecules, allowing for effective solubilization and separation in applications such as portable mass spectrometry. However, while PDMS can interact with polar molecules to some extent, its hydrophobic nature limits the strength and effectiveness of these interactions. For lighter polar molecules, PDMS can still provide some degree of separation, but for heavier polar molecules, strong interactions can lead to entrapment, disabling separation processes [40,41,42].

PEG, on the other hand, is hydrophilic due to its ether oxygen atoms, which can form hydrogen bonds with water and other polar molecules. This makes PEG highly compatible with polar substances, enabling interactions and effective separation of polar molecules [43,44]. So, in this work, we also considered how the addition of PEG chains to PDMS influences the interactions between bee alarm pheromones and polymer matrix by applying MD simulations. Finally, the simultaneous effects of temperature and the presence of PEG were also considered through MD simulations.

In this work, we considered a range of bee alarm pheromones, aiming to cover polar molecules with masses below and above 100 Da. The consideration of light polar pheromones enabled us to obtain information about the reference interaction energies that should be aimed for through modifications via temperature and/or PEG blending.

Last but not least, noncovalent interactions between polymers and bee pheromones were addressed via first-principles calculations, providing additional insights into how polymer blending can be utilized to make PDMS suitable for the separation of polar molecules of masses significantly higher than 100 Da.

## 2. Results and Discussion

### 2.1. Molecular Structures and Computational Workflow

Before referring to results, and due to the fact that a large number of systems were subjected to various calculations, we will first explain the molecular structures that were considered in this work. Regarding the MD simulations, our first task was to obtain interaction energies between pure PDMS and one pheromone molecule, and pure PEG and one pheromone molecule. This was achieved by generating an MD system consisting of 20 molecules in total, 19 polymer chains of PDMS or PEG, and one pheromone molecule. This type of MD system is illustrated in Figure 3.

The system presented in Figure 3 provided the best size-cost accuracy for our resources. PDMS chains are represented in the green ball-and-stick model, while furfural is shown in the van der Waals sphere model in dark brown. A list of pheromones is color-coded: Green designates important bee pheromones with a mass of less than 100 Da. Molecules marked yellow color are the so-called volatile disease markers (VDMs). The most important alarm bee pheromones for this work are marked in red in Figure 1, as these pheromones have a mass significantly higher than 100 Da. These are the pheromones for which we seek a separation solution for the MIMS system.

Polymer chains were prepared using the “Polymer Builder” tool of the Schrödinger Materials Science Suite. The number of monomer units used for generation of chains was set to 5. With optimized APB molecule polymer chains, MD systems were prepared using the “Disordered System Builder” tool of the Schrödinger Materials Science Suite. During the creation of the MD system, it was requested that the new cubic periodic system be built, while the initial state was requested to be Amorphous. Other settings of the tool for generating the MD system were a VdW scale factor of 0.5 and initial density of the component portion of the cell of 0.5. Placement attempts per molecule were set to 50, while attempts per density/VdW scale were set to 20.

The type of system presented in Figure 3 was first subjected to MD simulations at 300 K, aiming to collect information on interaction energies between each of the pheromone with a surrounding PDMS medium. Later, the MD simulations at different temperatures were performed for the case of pheromones for which the strongest interaction energy was obtained, to understand how temperature increase influences the strength of interaction.

The other type of MD system involved consisted of one pheromone molecule surrounded by PDMS chains and PEG chains, and this type of system is illustrated in Figure 4.

The system visualized in Figure 4 was used in this study to investigate the influence of PEG addition to PDMS on reducing the interaction between PDMS and a selected pheromone molecule. Specifically, we investigated the effect of presence of the PEG by adding one, two, and four PEG chains to the PDMS system. In all cases, the total number of polymer chains in our MD systems remained 19. Systems as described in Figure 3 were used for MD simulations at different temperatures to understand how this factor, together with the addition of PEG, influences the interaction between the selected pheromone and the PDMS/PEG blend. Systems, such as the one presented in Figure 4, were subjected to force-field-based MD simulations at different temperatures, specifically within the range of 300 K to 350 K. The lower limit of 300 K was chosen to represent room temperature, while the upper limit was selected based on the temperature range to which the MIMS device can be relatively easily heated.

After MD simulations, we considered two systems subjected to quantum-mechanical calculations, as presented in Figure 5. From now on, we will use the following color scheme for ball-and-stick molecular models: gray for carbon atoms, light gray for hydrogen atoms, orange for silicon atoms, and red for oxygen atoms.

These systems consisted of a PDMS chain and 11-eicosen (Figure 5a), while the other system consisted of one PEG chain and 11-eicosen (Figure 5b). The 11-eicosen pheromone was selected because the interaction energy was strongest between PDMS and this pheromone. Therefore, our task was to find ways to reduce the interaction energy between PDMS and 11-eicosen, which is representative of a heavy polar bee alarm pheromone.

The systems presented in Figure 5 were first subjected to geometrical optimizations using the GFN2-xtb method. This powerful semiempirical method was applied due to the size of the molecular systems, which consist of 183 and 133 atoms for the PDMS + 11-eicosen and PEG + 11-eicosen systems, respectively. Geometrically optimized systems were then subjected to single-point energy calculations at the M06-2X/6-31G(d,p) level of theory to obtain information about the formed noncovalent interactions using the Jaguar program of the Materials Science Suite. These optimized systems were also subjected to DFT calculations at the same level of theory, but with the ORCA5 modeling package, to generate the corresponding .wfn files that enabled us to create RDG scatter plots. Optimized ABP molecules have been summarized in Appendix A.

### 2.2. Interaction Energies via MD Simulations

In this section, we report interaction energies obtained between PDMS-based medium and the pheromone molecules listed in Figure 3. For this section, we performed 13 MD simulations at 300 K, and the obtained interaction energies are summarized in Figure 6.

Aside from interaction energies, the results presented in Figure 6 also contain information about the mass of the molecule (numbers at the top of each bin), which is important for identifying the interaction energies that indicate that PDMS is a good candidate as a potential sorbent. The results indicate that all light pheromones have interaction energies with PDMS of up to 30 kcal/mol. In particular, when studied light polar pheromones are taken into account, the highest interaction energy with PDMS is calculated for isobutyramide, equal to 29 kcal/mol. The two largest/heaviest pheromones, 11-eicosen and ethyl oleate, have much higher interaction energies of 51 and 38 kcal/mol, respectively.

Since it is known that PDMS is effective for the separation of light polar molecules up to 100 Da, the interaction energy of around 30 kcal/mol serves as a reference value. This reference is based on the fact that PDMS performs well with light polar molecules, and the highest interaction energy for these molecules is 30 kcal/mol, according to our results presented in Figure 6. Therefore, our goal is to reduce the interaction energy between PDMS and heavier polar molecules, such as 11-eicosen and ethyl oleate, to this reference value of around 30 kcal/mol.

### 2.3. Influence of Temperature and PEG Blending to Interaction Energies

The first method we used to reduce the interaction energies between PDMS and 11-eicosen was by increasing the temperature of the MD system consisting of 19 PDMS chains and one 11-eicosen molecule. In all cases, MD simulations were performed in the temperature range of 300 K to 350 K. The results of the interaction energy between the pure PDMS medium and the 11-eicosen molecule are summarized in Table 1.

The results presented in Table 1 clearly indicate that a simple increase in temperature leads to a significant decrease in the interaction energy (*E*_int_) between pure PDMS and 11-eicosen. Specifically, our calculations show that an increase of 50 K led to a 15% decrease in *E*_int_, from −51.52 kcal/mol to −44.11 kcal/mol. While this is a significant reduction in the magnitude of interaction energy, it is still much stronger than the reference value of −30 kcal/mol identified in the previous chapter.

The next method we tested for the reduction of the interaction energy was by adding PEG chains to our MD system consisting of PDMS chains and 11-eicosen. The obtained results for this case are summarized in Figure 7.

The results presented in Figure 7 show that the addition of PEG chains can also significantly influence the interaction energy with 11-eicosen. We studied the cases when one, two, and four PEG chains replaced one, two, and four PDMS chains, respectively. As shown in Figure 7, the presence of PEG chains significantly decreased the interaction energy *E*_int_ with the strongest effect observed when one PEG chain replaced one PDMS chain. In this particular case, the interaction energy between the polymer medium and 11-eicosen reduced to around −45.8 kcal/mol, which represents a decrease of more than 10%. Similar to the effect of increasing temperature, this reduction is notable but still significantly not good enough to reach the reference value of −30 kcal/mol.

Since both temperature increases and the presence of PEG chains produced positive effects in terms of reduction of the interaction energy between polymer medium and 11-eicosen, the logical next step was to study how interaction energy changes when both of these effects are taken into account. For these purposes, we performed an additional five MD simulations on a system where one PDMS chain is replaced by one PEG chain, in the temperature range between 300 K and 350 K, and calculated interaction energies. The obtained results are summarized in Figure 8.

As presented in Figure 8, the increase in temperature in the system with one PEG chain led to a significant decrease in the interaction energy between the polymer medium and 11-eicosen. Specifically, the temperature increase produced a much more notable effect when one PDMS chain was replaced by one PEG chain (orange curve in Figure 8), eventually leading to a decrease in $E_{int}$ to just −33.47 kcal/mol. This represents a substantial reduction of $E_{int}$ by 35%. More importantly, the value of −33.47 kcal/mol is very close to the previously established reference value of −30 kcal/mol. This indicates that the modification of the PDMS polymer medium with PEG, combined with a temperature increase, might be a promising approach for reducing the interaction energy and enabling the polymer medium to separate even heavy polar molecules such as 11-eicosen.

### 2.4. Binding Energies via First-Principles Calculations

The results obtained in previous sections provided valuable insights into how a combination of temperature increase and PEG blending can reduce the interaction energies between the polymer medium and 11-eicosen. For this pheromone, the aforementioned combination of parameters decreased the interaction energy from −51.52 kcal/mol to −33.47 kcal/mol, practically reaching the established reference value of around −30 kcal/mol. Since 11-eicosen has a significant mass of almost 300 Da, it is reasonable to expect that this combination of parameters would produce even better results for other polar molecules with a lower mass than 11-eicosen.

To gain deeper insights into interaction between polymer mediums and 11-eicosen, we have performed a combination of semiempirical and DFT calculations to study binding and formation of noncovalent interactions. The motivation for studying binding energies via quantum mechanical calculations is to gain insight into the preference of polymer chains for specific pheromone molecules. The results regarding binding energies obtained via the GFN2-xtb method are presented in Figure 9 and Figure 10.

Results regarding binding energies via quantum mechanical calculations revealed that, on average, PEG has a significantly higher preference toward considered pheromone molecules. Since we particularly were focused on 11-eicosen, we will now go into more detail according to obtained binding energies, namely, the $E_{b}$ in the case of interaction between PDMS and 11-eicosen is −14.40 kcal/mol, which is much lower than in the case of interaction between PEG and 11-eicosen, in which case the value of $E_{b}$ equal to −21.17 kcal/mol was calculated.

Thanks to this very large difference in preference of PDMS and PEG toward 11-eicosen, we are able to offer an explanation as to why the presence of PEG reduces the interaction energy. When PEG chains are introduced into the PDMS system, the stronger binding affinity of PEG for 11-eicosen plays a crucial role. Since PEG binds more strongly with 11-eicosen, it can effectively compete with PDMS for interaction with the pheromone. This competition results in a redistribution of interactions. Some of the 11-eicosen molecules that would have interacted with PDMS now interact with PEG instead. Furthermore, the overall interaction energy between the polymer medium (a mix of PDMS and PEG) and 11-eicosen is reduced because PEG’s stronger binding helps “pull” 11-eicosen away from PDMS, leading to a decrease in the interaction energy of the system as a whole.

As for the combined effect with temperature, the reduction in interaction energy observed when the temperature is increased in the presence of PEG chains can be explained by the increased molecular motion at higher temperatures. This increased motion allows for more effective interactions between PEG and 11-eicosen, further enhancing the reduction in interaction energy.

### 2.5. Noncovalent Interactions

Since the stronger preference of one polymer in this case plays a critical role in enabling PDMS to be used for heavier polar molecules such as 11-eicosen, we were motivated to further investigate the noncovalent interactions formed between PDMS and 11-eicosen, and PEG and 11-eicosen. For this purpose, we first performed an analysis of electron density using the Jaguar program, which helped reveal the bond critical points and noncovalent interactions with their corresponding strengths. GFN2-xtb-optimized complexes of PDMS-11-eicosen and PEG-11-eicosen, along with noncovalent interactions obtained via the M06-2X/6-31G(d,p) level of theory, are presented in Figure 11.

The very strong binding energies obtained with the GFN2-xtb method for the interaction between PDMS/PEG and 11-eicosen were expected due to the high number of noncovalent interactions formed between these polymer chains and the mentioned molecule. Specifically, the results in Figure 11 clearly explain why a much stronger binding energy was calculated for PEG. Simple counting of the number of noncovalent interactions shows that in the case of PEG, six more noncovalent interactions were formed compared to when PDMS interacts with 11-eicosen. Additionally, it can be seen from Figure 11 that the distribution of bond critical points in the case of PEG is practically evenly distributed along the line of interaction with 11-eicosen, whereas the central part of PDMS lacks bond critical points and noncovalent interactions compared to PEG.

## 3. Computational Details

In this work, a set of MD simulations were performed with the Desmond [45] program, as implemented in the Schrödinger Materials Science Suite (SMSS), version 2022-3 [46,47,48,49]. For all MD simulations with the Desmond program, the parameters included the OPLS3e force field [50,51,52,53], with the simulation time of 20 ns; NPT ensemble of particles; and normal pressure.

Geometrical pre-optimizations of alarm bee pheromone (ABP) molecules, to obtain reactive properties, were performed using the GFN2-xTB method, developed by Prof. Grimme and coworkers [24,54,55,56]. Obtained structures were re-optimized at the DFT level, utilizing the B3LYP [57,58,59,60] functional and 6-31G(d,p) basis set. Finally, to obtain information about quantum molecular and reactivity descriptors, optimized molecular structures were subjected to single-point energy calculations with the M06-2X functional [61,62,63,64] and 6-311++G(d,p) basis set.

Geometrical optimizations of systems consisting of two polymer chains or of polymer plus ABP molecule were performed using the GFN2-xTB method. These systems and calculations were used to obtain information about the binding energies according to the following equation:(1)$$E_{b}=E_{tot}\left(P+ABP\right)-E\left(P\right)-E\left(ABP\right)$$
where *E*_tot_ (P + ABP) denotes the total energy of the optimized complex consisting of a polymer and ABP molecule, and *E*(P) denotes the total energy of the optimized polymer chain, while *E*(ABP) denotes the total energy of an ABP molecule.

GFN2-xTB calculations were performed as implemented in the xtb 6.5.1. program. GFN2-xTB calculations were run within the atomistica.online molecular modeling platform [65,66], freely available at https://atomistica.online. DFT calculations were performed with the Jaguar program [46,67,68,69], as implemented in the Schrödinger Materials Science Suite 2022-3. The Jaguar program was used to identify and quantify noncovalent interactions.

## 4. Conclusions

In this study, we have explored the molecular interactions between bee alarm pheromones and polymer materials such as PDMS and PEG to enable the efficient separation of these pheromones in MIMS systems. The primary goal was to reduce the strong interactions between heavier polar bee alarm pheromones and the PDMS polymer to enhance its applicability as a sorbent material for gas separation systems.

Initial MD simulations showed that pure PDMS exhibits strong interactions with heavier polar pheromones like 11-eicosen and ethyl oleate, with interaction energies of −51.52 kcal/mol and −38 kcal/mol, respectively. These values are significantly higher than the reference interaction energy of around −30 kcal/mol for lighter polar molecules, indicating the need for modification of PDMS to reduce these interactions.

Increasing the temperature of the MD system from 300 K to 350 K led to a 15% decrease in the interaction energy between PDMS and 11-eicosen, from −51.52 kcal/mol to −44.11 kcal/mol. While this reduction is significant, it is not sufficient to meet the reference value.

Adding PEG chains to the PDMS system notably reduced the interaction energy between the polymer medium and 11-eicosen. The replacement of one PDMS chain with one PEG chain resulted in an interaction energy of −45.8 kcal/mol, demonstrating a more than 10% reduction. However, this alone was still insufficient to achieve the desired reference value.

When both temperature increase and PEG blending were applied, the interaction energy between the polymer medium and 11-eicosen decreased significantly. At 350 K, with one PEG chain added, the interaction energy dropped to −33.47 kcal/mol, approaching the target of −30 kcal/mol. This indicates that the combination of PEG blending and temperature adjustment is a promising strategy for reducing interaction energies and enhancing the separation efficiency of PDMS for heavier polar molecules.

PEG’s stronger binding preference for pheromones, as evidenced by the binding energy calculations, means that it can outcompete PDMS in interacting with the pheromones. This results in a redistribution of interactions where some pheromone molecules that would have been strongly bound to PDMS now bind to PEG instead. This competition and redistribution lead to an overall reduction in the interaction energy of the system.

Quantum mechanical calculations further revealed that PEG has a higher binding preference for 11-eicosen compared to PDMS. The binding energy of PEG with 11-eicosen was found to be −21.17 kcal/mol, significantly stronger than the −14.40 kcal/mol for PDMS. This higher affinity of PEG helps to reduce the overall interaction energy when blended with PDMS.

Analysis of noncovalent interactions showed that PEG forms more numerous and evenly distributed interactions with 11-eicosen compared to PDMS. This explains the stronger binding affinity of PEG and its effectiveness in reducing the interaction energy when blended with PDMS.

## Figures and Tables

**Figure 1 ijms-25-08599-f001:**
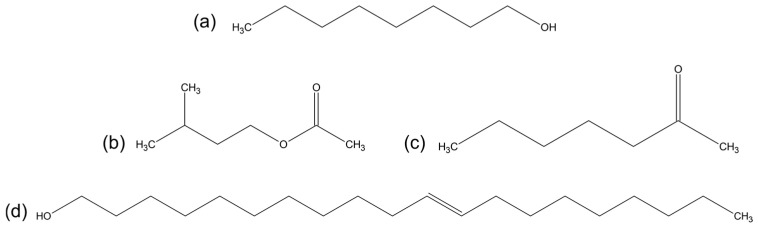
Structures of selected well-known alarm pheromones: (**a**) 1-octanol, (**b**) isopentyl acetate, (**c**) 2-heptanone, and (**d**) (Z)-11-Eicosane-1-ol.

**Figure 2 ijms-25-08599-f002:**
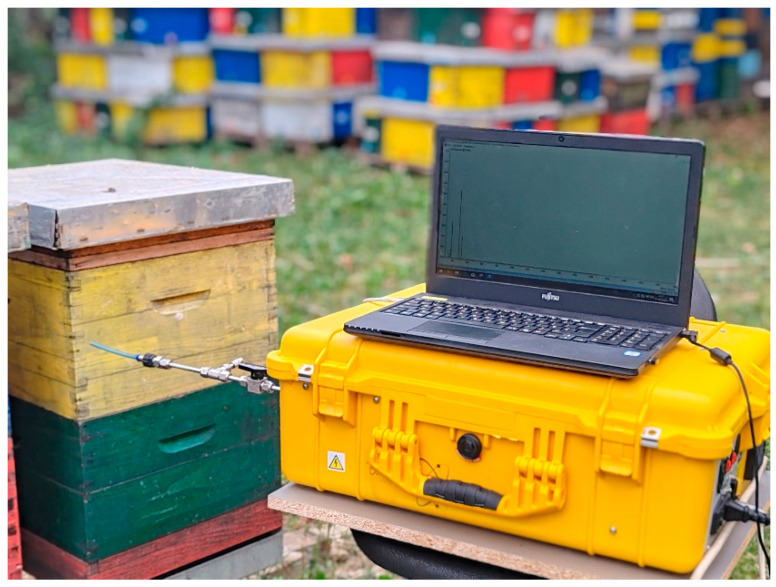
Realized MIMS system. Reprinted from ref. [12].

**Figure 3 ijms-25-08599-f003:**
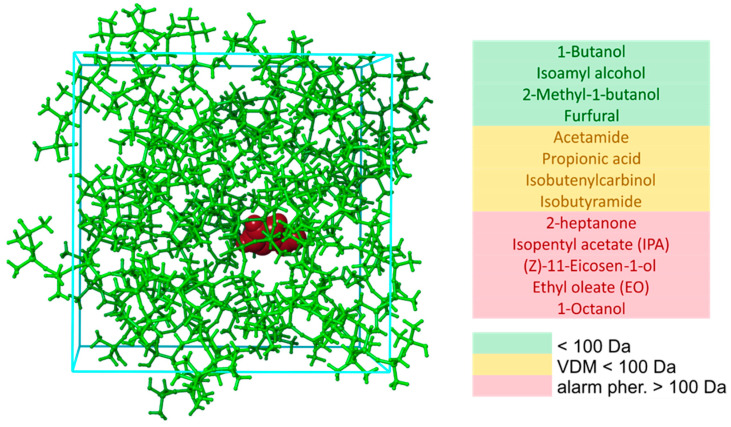
MD system consisting of 19 PDMS chains (green color), one molecule of furfural (brown color), and a list of studied pheromones.

**Figure 4 ijms-25-08599-f004:**
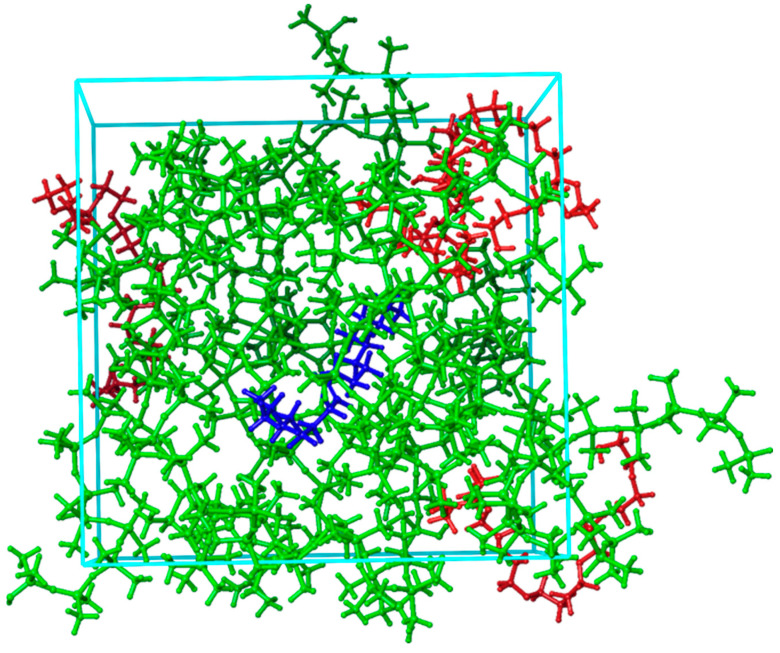
MD system: 11-eicosen pheromone (blue-colored molecule) surrounded by 15 PDMS (green-colored molecules) and 4 PEG (red-colored molecules) chains.

**Figure 5 ijms-25-08599-f005:**
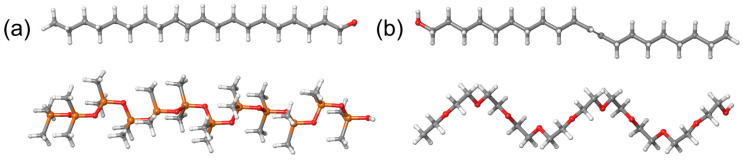
Starting molecular configurations of (**a**) PDMS + 11-eicosen and (**b**) PEG + 11-eicosen.

**Figure 6 ijms-25-08599-f006:**
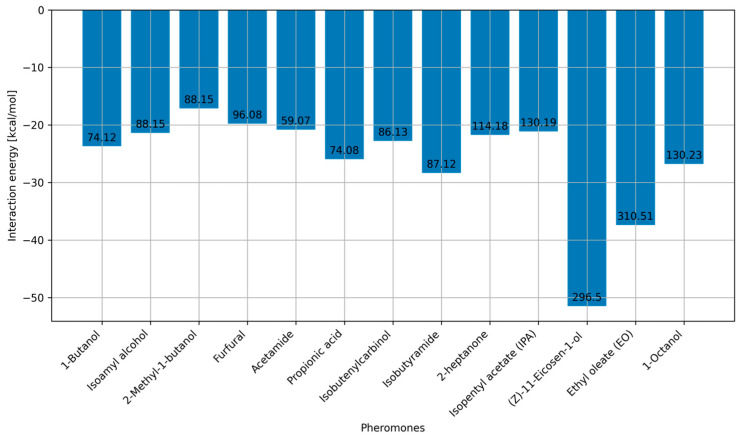
MD interaction energies between pure PDMS and pheromones.

**Figure 7 ijms-25-08599-f007:**
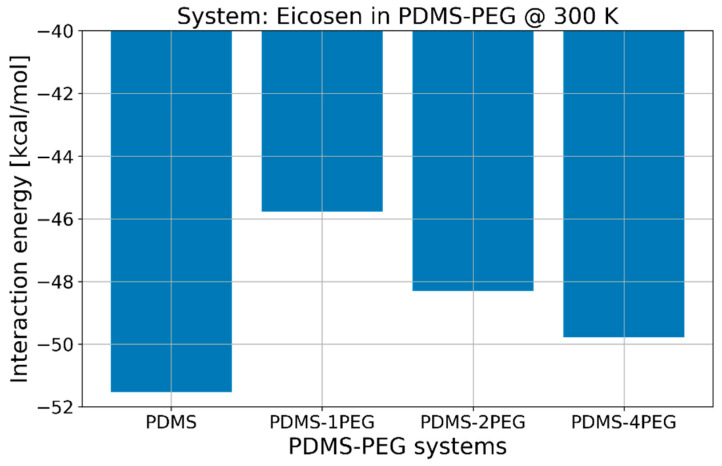
Influence of PEG addition to interaction energies with 11-eicosen at 300 K.

**Figure 8 ijms-25-08599-f008:**
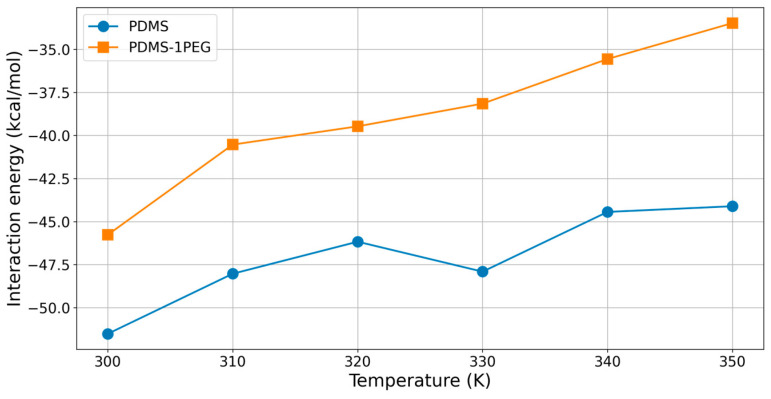
Interaction energy of two polymer mediums (PDMS and PDMS-1PEG) with 11-eicosen.

**Figure 9 ijms-25-08599-f009:**
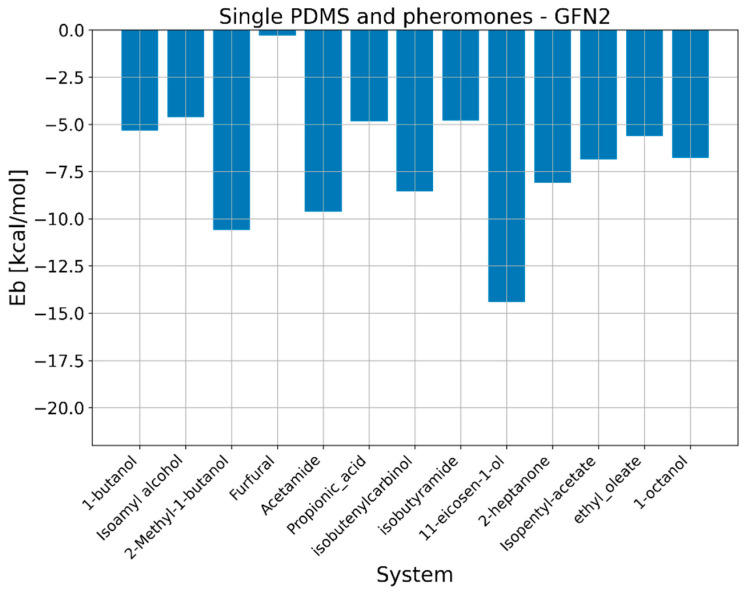
GFN2-xtb binding energies between PDMS and pheromone molecules.

**Figure 10 ijms-25-08599-f010:**
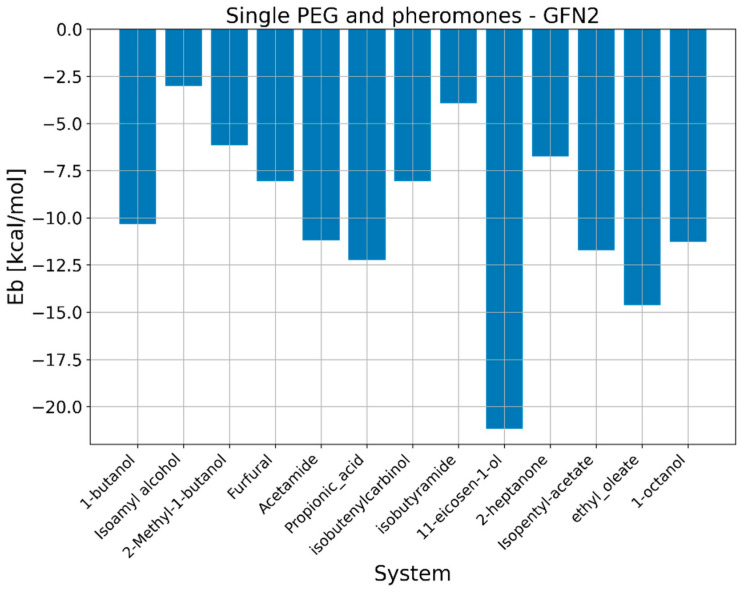
GFN2-xtb binding energies between PEG and pheromone molecules.

**Figure 11 ijms-25-08599-f011:**
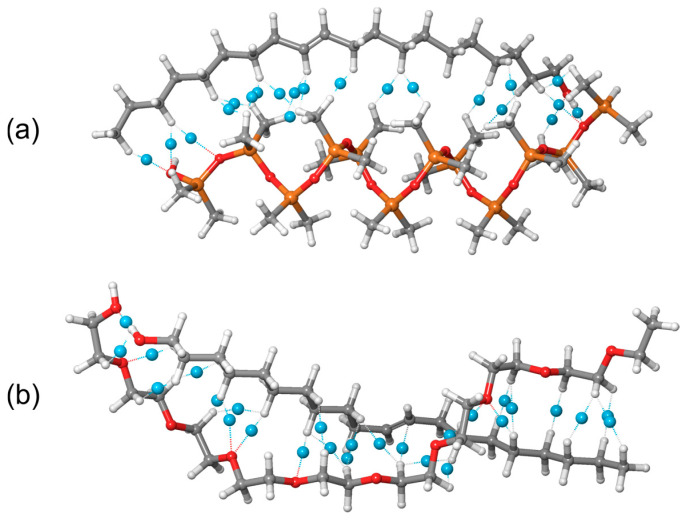
Visualization of bond critical points (blue dummy atoms) and corresponding noncovalent interactions. (**a**) PDMS + 11-eicosen and (**b**) PEG + 11-eicosen systems.

**Table 1 ijms-25-08599-t001:** Interaction energies between PDMS medium and 11-eicosen at different temperatures.

Temperature [K]	*E*_int_ [kcal/mol]
300	−51.52
310	−48.03
320	−46.17
330	−47.91
340	−44.44
350	−44.11

## Data Availability

The original contributions presented in the study are included in the article’s materials, and further inquiries can be directed to the corresponding author.

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
