# Peer review of "Design of Novel Membranes for the Efficient Separation of Bee Alarm Pheromones in Portable Membrane Inlet Mass Spectrometric Systems"

_ijms, 2024, doi:10.3390/ijms25168599_

Round 1

Reviewer 1 Report

Comments and Suggestions for Authors

In the submitted, purely computational, manuscript the Authors have studied several (12) small organic molecules acting as bee pheromones with chosen (2) molecules imitating the polymers (PEG, PDMS). While I appreciate the novelty of such approach, the work needs significant revisions. My detailed comments can be found below.

Also, the results should be somehow connected and discussed with relation to previously published experimental results.

Major comments:

Lines 59-66, the figure presenting the chemical structures of those compounds should be included in the introduction

Line 90, here, please present the picture of MIMS, to increase the clarity. You can copy the image from the other article, just please remember about the copyright license agreement.

Works [15-30], why those particular works are being cited? They are very loosely connected with the current topic. Instead, I’d recommend to cite some recent review works on the applications of DFT and MD in various systems.

Lines 361-632, MD simulations are usually run in triplicate to increase the credibility of results. Has it been done in the current work as well?

Line 363, the list of the optimized ABP should be included here or in the supplementary information file

Lines 174-180, It is not described how the periodic boxes for the Desmond MD runs were created. I mean, what was the assumed density? How the initial conformations were generated? Those are quite large molecules, it’s not water that can be added randomly. The choice of the initial structure may have a great influence on the received results. Besides, how the initial position of the pheromone within this system was set? Randomly? This must be stated clearly.

Line 205, why haven’t the Authors gone above 4? Let’s say 10:9 molar ratio? Or even pure PEG to see the difference. In my opinion, this must be updated, at least for one chosen system.

I know the Authors did they best to present the results in a clear way but it is still not very clear to me and, potentially, to other readers. Correct me if I’m wrong. The large system (20+1 molecules, periodic one) was subjected solely to Molecular Mechanics calculations? Then, the semiempirical ones, they’ve been done on single molecules? Not on this large system?

Why the values of DFT energies have not been presented in a form of a figure/table? Also, why the Authors haven’t calculated the free energies?

Why the results of Isoamyl alcohol are not presented at all, despite it being listed in Figure 1?

Author Response

Dear Reviewer,

On behalf of all co-authors, I would like to sincerely thank you for your time and expertise in thoroughly reviewing our manuscript. We found your comments to be constructive and have addressed them all in the report provided below. All changes in the manuscript have been marked with yellow color.

Thank you once again.

Comment 1:

Also, the results should be somehow connected and discussed with relation to previously published experimental results.

Reply:

Thank you for your observation. Additional comments have been added to the Introduction section to explain the connection between our results and previously published experimental results, citing our previous relevant papers. The importance of making modifications to the PDMS membrane to increase the range of molecules that we can detect using MIMS has also been mentioned in the Introduction. In the revised manuscript, these additions are in lines 122 – 132 and 139 – 146.

Major comments:

Comment 2:

Lines 59-66, the figure presenting the chemical structures of those compounds should be included in the introduction

Reply:

Thank you for suggesting this. We have added the structural formulas of the compounds mentioned in the corresponding paragraph (Figure 1 of the revised manuscript)

Comment 3:

Line 90, here, please present the picture of MIMS, to increase the clarity. You can copy the image from the other article, just please remember about the copyright license agreement.

Reply:

Thank you for suggesting this. A picture of our MIMS system has been added (Figure 2 of the revised manuscript).

 Comment 4:

Works [15-30], why those particular works are being cited? They are very loosely connected with the current topic. Instead, I’d recommend to cite some recent review works on the applications of DFT and MD in various systems.

Reply:

In this paragraph, we aim to highlight the power of computational methods, such as DFT calculations and MD simulations, in understanding the various properties of molecules. In our papers, we usually demonstrate, in one or two paragraphs in the Introduction, how different computational methods have been applied to understand various structural and reactive properties of molecules. In these paragraphs, we also tend to reference some of our papers to emphasize our demonstrated history in applying these methods to investigate various types of molecular structures.

Additionally, we completely agree with your suggestion that such paragraphs should include citations from review papers, providing readers with valuable sources for further information on the usability of these methods. Therefore, we have expanded this paragraph and added citations relevant review papers (References 17 – 29 of the revised manuscript, while the newly added text is in lines 148 - 153).

Comment 5:

Lines 361-632, MD simulations are usually run in triplicate to increase the credibility of results. Has it been done in the current work as well?

Reply:

We entirely agree with you about the necessity of running MD simulations in triplicates. As a matter of precaution, in our studies we always choose one representative system and perform three MD simulations on it using the same set of parameters and the same starting configuration. We then analyze the obtained results (visualize total energy vs time (steps) and temperature vs time (steps) to ensure that the simulations are stable). In the case of this study, we treated the system comprising of a polymer medium and 11-eicosen molecule to be our representative system because we wanted to check if it is possible to reduce the interaction energy to those values corresponding to interaction energy between polymer medium and ABP molecules of low mass. Therefore, three MD simulations have been performed on a system consisting of PDMS chains and 11-eicosen molecule. As mentioned previously, the results of these simulations have been analyzed and we have concluded that the selected simulation parameters mentioned in the computational details section provided stable simulations.

Comment 6:

Line 363, the list of the optimized ABP should be included here or in the supplementary information file

Reply:

Thank you very much for this suggestion. We have prepared a supplementary file that contains the list of optimized structures together with their optimized geometries.

Comment 7:

Lines 174-180, It is not described how the periodic boxes for the Desmond MD runs were created. I mean, what was the assumed density? How the initial conformations were generated? Those are quite large molecules, it’s not water that can be added randomly. The choice of the initial structure may have a great influence on the received results. Besides, how the initial position of the pheromone within this system was set? Randomly? This must be stated clearly.

Reply:

Thank you for mentioning this, as the previous version of the manuscript didn’t contain any information on how the systems for MD simulations were prepared. In the revised version of the manuscript, we have enriched the corresponding section with one more paragraph explaining the details of how MD systems were generated. In general, we used the built-in tool of the Schrödinger Materials Science Suite to generate systems for MD simulations. The most important parameters of this procedure are added in the chapter 2.1. We also added information about how polymer chains were obtained. The newly added text is located in lines 223-231.

Comment 8:

Line 205, why haven’t the Authors gone above 4? Let’s say 10:9 molar ratio? Or even pure PEG to see the difference. In my opinion, this must be updated, at least for one chosen system.

Reply:

We are very glad you mentioned this and we will gladly elaborate on why we decided to study the particular systems from our manuscript. The reason for not considering more than four PEG chains is due to its hydrophilic properties, which can endanger the high vacuum required for the safe operation of our portable membrane inlet mass spectrometer (MIMS). Our MIMS instrument has an operational pressure of around 1x10-6 Torr. To hold a vacuum, we require predominant hydrophobic membrane material with flexible surface area, chemical composition and thickness, such as PDMS. In addition, an evaluation of the benefits of different hydrophobic membrane materials for MIMS operation and VOC analysis has been described by R.C. Johnson et al., JMS, 1997 (https://doi.org/10.1002/(SICI)1096-9888(199712)32:12%3C1299::AID-JMS589%3E3.0.CO;2-P). Additionally, even with two PEG chains, we didn’t notice improvement, but we wanted to test, just in case, the systems containing four PEG chains, so we already went behind upper limits.

Comment 9:

I know the Authors did they best to present the results in a clear way but it is still not very clear to me and, potentially, to other readers. Correct me if I’m wrong. The large system (20+1 molecules, periodic one) was subjected solely to Molecular Mechanics calculations? Then, the semiempirical ones, they’ve been done on single molecules? Not on this large system?

Reply:

We appreciate your concern and are thankful for your understanding. Indeed, visualizing systems with dozens of large molecules is challenging. That is why we have included additional text in the caption of Figure 4, to indicate which collor corresponds to which molecule clearly.

Yes, you understood correctly. The large systems, such as the one presented in Figure 3, were subjected to force-field-based MD simulations at different temperatures. In contrast, the systems presented in Figures 5a and 5b underwent semiempirical GFN2-xTB calculations. Given the size of the systems in Figure 5, we opted for the modern semiempirical approach developed by Prof. Grimme and colleagues. While we acknowledge that this approach may not provide energies as accurate as DFT or some wavefunction methods, literature surveys indicate that these semiempirical methods are reliable, especially for organic structures like those studied in this work.

Comment 10:

Why the values of DFT energies have not been presented in a form of a figure/table? Also, why the Authors haven’t calculated the free energies?

Reply:

DFT energies haven’t been presented since we didn’t perform DFT calculations on all structures. DFT calculations have been performed only on APB molecules, after pre-optimizations at GFN2-xtb level of theory. Due to the size of the systems and the flexibility of polymer chains, we decided to perform optimizations between single polymer chain and selected APB using the GFN2-xtb level of theory. While we are aware of the limitations related to semiempirical calculations versus the DFT calculations, it was the size of the molecular systems that motivated us to employ a semiempirical method in order to obtain binding energies.

Comment 11:

Why the results of Isoamyl alcohol are not presented at all, despite it being listed in Figure 1?

Reply:

Thank you so much for noticing this. We accidentally forgot to calculate GFN2 binding energies for the case of isoamyl alcohol. For the revised version of our manuscript, we performed these calculations as well and updated corresponding figures (now Figures 9 and 10).

Thank you once again.

All the best,
Stevan Armaković

Reviewer 2 Report

Comments and Suggestions for Authors

The manuscript of Armaković et al reports the use of multilevel modeling techniques to understand molecular interactions between representative bee alarm pheromones and polymers such as polymethyl siloxane (PDMS), polyethylene glycol (PEG) and their blend. Thes studies were carried out applying powerful computational atomistic methods based on a combination of modern semiempirical (GFN2-xtb), first principles (DFT), and force field calculations.

In my opinion this study is interesting although to allow its publication it needs to be improved in some points listed below:

1.      The authors should provide information on the software used (the operating principles). Could you imagine a specific paragraph? In this way, information could also be given on the set ups used for the reproducibility of the data

2.      What temperatures range are used in addition to 300 K for the preliminary study on system used? (see paragraph 2.1)

Author Response

Dear Reviewer,

We are using this opportunity to sincerely thank you for your time and expertise, and your positive feedback. We have carefully studied your comments and made the required revisions. Bellow, we are reporting the changes that have been made. All changes in the manuscript have been marked with yellow color.

Comment 1:

The authors should provide information on the software used (the operating principles). Could you imagine a specific paragraph? In this way, information could also be given on the set ups used for the reproducibility of the data

Reply:

Thank you for mentioning this. In the revised version of manuscript, we have added explanations in the sub-chapter 2.1. to make the reproducibility of the results easier. In particular, we explained in more details how MD systems were built.

Comment 2:

What temperatures range are used in addition to 300 K for the preliminary study on system used? (see paragraph 2.1)

Reply:

Thank you very much for mentioning this, since the choice of temperature is very important. Later, in sub-chapter 2.3 we have mentioned the temperature range, but this information should be also present in chapter 2.1. Additionally, we forgot to shortly elaborate why we selected the mentioned temperature range. The choice of temperatures was made by taking into account the temperature range that can be covered by our practically realized MIMS device. Currently, our MIMS device can be heated up to around 75 °C, because of which we set temperature range for our MD simulations to be between the room temperature of 300 K to 350 K. MD simulations were performed in temperature steps of 10 K.

Thank you once again.

All the best,
Stevan Armaković

Round 2

Reviewer 1 Report

Comments and Suggestions for Authors

The Authors have revised and improved their manuscript. In its current form, it can be accepted.